# A Bi-Objective Approach to Minimize Makespan and Energy Consumption in Flow Shops with Peak Demand Constraint

**Weiwei Cui** [1] 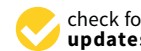 **and Biao Lu** [2],*

1  School of Management, Shanghai University, Shanghai 200444, China; cuiww67@163.com
2  College of Economic and Management, Nanjing University of Aeronautics and Astronautics, Nanjing 211106, China
*  Correspondence: lubiao123@nuaa.edu.cn

**Abstract:** With the growing concern of energy shortage and environment pollution, the energy aware operation management problem has emerged as a hot topic in industrial engineering recently. An integrated model consisting of production scheduling, preventive maintenance (PM) planning, and energy controlling is established for the flow shops with the PM constraint and peak demand constraint. The machine's on/off and the speed level selection are considered to save the energy consumption in this problem. To minimize the makespan and the total energy consumption simultaneously, a multi-objective algorithm founded on *NSGA-II* is designed to solve the model effectively. The key decision variables are coded into the chromosome, while the others are obtained heuristically using the proposed decoding method when evaluating the chromosome. Numerical experiments were conducted to validate the effectiveness and efficiency by comparing the proposed algorithm and the traditional rules in manufacturing plant. The impacts of constraints on the Pareto frontier are also shown when analyzing the tradeoff between two objectives, which can be used to explicitly assess the energy consumption.

**Keywords:** flow shop; makespan; energy consumption; peak demand

## 1. Introduction

The effectiveness of operations management is crucial for improving the profit of manufacturing company in the fierce market competition. Thus, the flow shop scheduling problem has already been investigated for many years, which is not only a hot topic of researchers but also an interested issue of practitioners in industry. The related research results can be found in the articles and review papers, such as those of Ribas et al. [1], Yenisey et al. [2], and Komaki et al. [3]. Although different kinds of assumptions and details are considered in the literature, it is still important to gather information and explore the features of manufacturing plant to fill the gap between theory and reality since the world changes so quickly in the 21st century.

According to the International Energy Outlook by the U.S. Energy Information Administration (USEIA, 2016), the total world energy consumption is projected to increase from 549 quadrillion British thermal units (Btu) in 2012 to 815 quadrillion Btu in 2040 [4]. It means that a 48% increase from 2012 to 2040 will further aggravate the situation of energy shortage and environment pollution. Thus, sustainability and green economy are attracting more and more attention from the researchers and practitioners in different fields recently. It is reported that 54% of the world energy consumption is caused by the industrial sector [5]. Furthermore, 90% of energy consumption and 84% of $CO_2$ emission in the industrial sector are attributed to the manufacturing activities [6]. All human beings have reached

a consensus that improving the energy efficiency of production systems, i.e., green manufacturing, is a good method and also a necessary road towards a cleaner future. Therefore, energy-related objectives are added into the consideration when researchers try to optimize the traditional economic objectives [7].

The mixture of economic target and environmental impact leads to multi-objective approaches. Some researchers design a weighted sum of all objectives and use the single-objective algorithm to solve the model [8]. Some researchers search the Pareto solutions using multi-objective algorithms [9]. No matter which kind of method is adopted, it is inevitable that the complexity of production scheduling problem with energy consideration is higher than the traditional problems. Firstly, the tradeoff between two objectives usually leads to a conflict in decision making. Secondly, the discussion of several energy-saving techniques also increases the categories of decision variables, such as shutting down machine to save the energy consumption during idle time, selecting a lower speed level to reduce the power demand of machine tools. Thirdly, inserting idle time into the plan appropriately may improve the system performance by postponing the processing of jobs, which adds the continuous variables into the model including many discrete variables.

In addition, some new constraints must be added into the traditional flow shop scheduling model in order to accord with the realistic circumstance in work shop. Two significant factors are peak demand constraint and preventive maintenance (PM) constraint. Chupka et al. [10] stated that investing 2 trillion dollars will be required to build the facilities to satisfy the booming demand by 2030. Since most of power demand occurs in the peak hours, which only occupy 1% of the year, many facilities are left idle most of the time. Thus, the manufacturing plant is required to reduce the maximum power demand and keep it below the peak value. Cui et al. [11] stated that PMs need to be performed periodically to guarantee the reliability of machine during the production horizon. Thus, the machine must be turned off when performing a PM, which means it cannot process the job at the same time. The existence of constraints increases the difficulty of solving the combinatorial optimization problems by defining the shape of feasible space. If production, maintenance, and energy decisions are determined independently by their own departments, there is a high likelihood of infeasibility.

The contribution of this paper is stated as follows: An integrated model with PM constraint and peak demand constraint is established for related departments to coordinate the production scheduling, PM planning, and energy controlling in the manufacturing plant. Meanwhile, an effective algorithm is designed to solve the model within an acceptable computation time for the large-sized problems in reality. The research results can provide a whole solution guiding the managers to minimize cost and maximize profit when fulfilling the duty in the sustainability.

The remainder of this paper is organized as follows. A brief literature review is provided in Section 2. Section 3 describes the problem in detail with the mathematical programming model. Then, a combination of *NSGA-II* framework and decoding method is designed in Section 4 to find the Pareto frontiers. Section 5 validates the effectiveness and the efficiency of the proposed algorithm in the numerical experiments. The conclusions and future work are shown in Section 6.

## 2. Literature Review

Many articles focus on the flow shop system in the production field. From the viewpoint of scheduling, different variants of this problem are studied. For example, the blocking flow shop problem is proposed in [12], the no-wait flow shop problem is proposed in [13], the sequence-dependent set-up times flow shop problem is discussed in [14], etc. We only briefly review the literature closely related to our research from two directions: the flow shop with PMs and the flow shop with energy consideration.

Performing PM causes the unavailability of machine since no job can be processed on the machine at the same time. Two kinds of assumptions are investigated by the researchers: (1) the unavailable intervals are known and fixed in advance; and (2) the unavailable intervals are flexible and scheduled by the manager. In the first assumption, some researchers considered one maintenance period in the horizon, for example the authors of [15–17] focused on scheduling resumable jobs in two-machine flow

shops and the authors of [18,19] focused on the non-resumable jobs. Some researchers considered that the intervals are periodically fixed, for example the authors of [20–22] investigated the two-machine flow shops and the authors of [23,24] studied the multi-machine flow shops. In the second assumption, some researchers considered that the continuous working time of machine must be smaller than a threshold, for example the authors of [25] compared the difference between fixed constraint and flexible constraint. Some researchers considered that the PM must be performed in a predefined interval, for example the authors of [26] developed a hybrid genetic algorithm with tabu search and the authors of [27] developed an artificial immune algorithm to solve the problem.

The energy objective added into the problem usually requires the multi-objective models and algorithms. The authors of [28] adopted the Non-dominant Sorting Genetic Algorithm (NSGA) to solve the bi-objective of the energy consumption and the total weighted tardiness, which aims to minimize the non-processing energy through reducing the machine's idle time for the job shops. The authors of [29] analyzed the optimal cutting parameters of machine tools and determined the jobs' sequence of flexible flow shop by combining the two objectives into one objective function. The authors of [30] considered the setup energy consumptions and designed a multi-objective algorithm to solve the hybrid flow shop problems. The authors of [31,32] considered the energy consumption of AGV transporting jobs and designed hybrid meta-heuristics to minimize the makespan and energy consumption of flexible job shops. The above references intend to reduce the energy consumption in the original framework of scheduling problem, while the other references intend to discuss how to use the energy-saving techniques.

Shutting down idle machine is an effective technique to reduce the energy consumption. The authors of [33–35] designed a greedy randomized multi-objective adaptive search metaheuristic, a non-dominated sorting genetic algorithm II, and a $\varepsilon$-constraint method to reduce the non-processing energy using the power-down mechanism for a single machine system, respectively. The authors of [36,37] developed a teaching-learning-based optimization algorithm and a hybrid multi-objective backtracking search algorithm to solve the similar problem for the flow shops. The authors of [38] developed a multi-objective genetic algorithm based on *NSGA-II* to solve the problem for the job shops to minimize the total weighted tardiness and total non-processing energy consumption. The authors of [39] considered energy consumption of transmission belt between consecutive machines in flow shops and also adopted the power-down mechanism to save the energy consumption.

The power demand of machine is larger when the speed level of tool is higher. Thus, some researchers intend to use the lower speed to reduce the energy consumption with the sacrifice of longer processing time. The authors of [40,41] investigated this problem for the two-machine flow shops and multi-machine flow shops, respectively. The authors of [42] designed a multi-objective genetic algorithm with two refinement strategies based on local search for the job shops. The authors of [43] designed a teaching-learning-based algorithm to tackle the hybrid flow shop problem, which consists of three aspects including task assignment, job sequencing, and speed selecting. The authors of [44] adopted a chance-constrain approach to describe decision-makers' awareness for the total tardiness when minimizing the bi-objective of makespan and energy consumption. The authors of [45] considered the makespan, tardiness, and energy consumption and assumed that the third objective is less important than other ones. The authors of [46] proposed an adaptive multi-objective variable neighborhood search algorithm to solve the no-wait flow shop problem, and the authors of [47] designed a multi-objective grey wolf optimization algorithm to solve the flexible job shop problem. The authors of [48] studied the flexible job shop scheduling problem considering the machines' on/off and speed level simultaneously.

Peak demand is not considered in the above literature. However, it plays an important role in the energy cost of manufacturing plant and the burden of power utility. The authors of [49] inserted idle time into the scheduling to minimize the sum of peak power cost and inventory cost. The authors of [50] used the discrete event simulation to formulate the system's peak load. The authors of [51] studied the Bernoulli production line to minimize the peak demand cost and labor cost. The authors

of [52] considered the heating ventilation and air conditioning system to reduce the peak load of manufacturing system. The authors of [51,52] focused on the tact-system production rather than jobs sequencing, and they considered corrective maintenance for random breakdowns rather than PM. The authors of [53] is the first attempt to tackle the permutation flow shop scheduling problem with peak demand constraint, in which the mixed integer programming formulations are provided to minimize the single objective of makespan. Then, the authors of [54] is the first attempt to solve this problem using meta-heuristics with effective decoding methods. However, they only considered the speed level and ignored the impact of PMs and machine's on/off.

In summary, the management of flow shop is a multifaceted issue related to the production requirements, machine's maintenance, and energy factors. The articles in the literature focus on different aspects to improve the efficiency from different ways. The purpose of our paper is to establish an integrated model to minimize the bi-objective of makespan and energy consumption in flow shops with PM constraint and peak demand constraint. Compared with the methods in [12–27], this paper belongs to the energy-efficient scheduling problem. Compared with the methods in [28–48], the main difference of this paper is considering the impact of PM constraint and peak demand constraint. Compared with the methods in [49–52], this paper focuses on the flow shop scheduling area and treat the peak demand as a hard constraint rather than an objective. Compared with the methods in [53,54], this paper tries to save the energy consumption by turning off the idle machine and considers the fact that machines must be maintained periodically to keep a high reliability. A summary of literature review can be found in Table 1, which showd the differences between different references.

**Table 1.** A summary of literature review.

| Reference | Production Scheduling | | Maintenance Planning | | | Energy Controlling | | |
|---|---|---|---|---|---|---|---|---|
| | Production | Jobs Sequencing | Fixed | Flexible | Objective | On/Off | Speed Selection | Peak Demand |
| [12–14] | √ | √ | × | × | × | × | × | × |
| [15–24] | √ | √ | √ | × | × | × | × | × |
| [25–27] | √ | √ | × | √ | × | × | × | × |
| [28–32] | √ | √ | × | × | √ | × | × | × |
| [33–39] | √ | √ | × | × | √ | √ | × | × |
| [40–47] | √ | √ | × | × | √ | × | √ | × |
| [48] | √ | √ | × | × | √ | √ | √ | × |
| [49] | √ | √ | × | × | √ | × | × | √ |
| [50] | √ | × | × | × | √ | × | √ | √ |
| [51,52] | √ | × | × | × | √ | √ | × | √ |
| [53,54] | √ | √ | × | × | √ | × | √ | √ |
| This paper | √ | √ | × | √ | √ | √ | √ | √ |

## 3. Problem Statement

### 3.1. Problem Description

In this paper, we study a flow shop composed of several machines $M = \{M_1, M_2, \ldots, M_m\}$. A set of jobs $J = \{J_1, J_2, \ldots, J_n\}$ needs to be processed from $M_1$ to $M_m$. Thus, each job $J_i$ includes a sequence of operations $\{O_{ij}\}$. The basic processing time of $O_{ij}$ on machine $M_j$ is denoted by $p_{ij}^0$. All jobs are ready to be processed at time zero. The completion time of the last job is denoted by $C_{max}$, which equals to its finish time on the last machine.

PMs need to be performed periodically for each machine to guarantee a high reliability. Maintenance time of $M_j$ equals to $pt_j$. When a PM is performed on a machine, it cannot process the job at the same time. The reason is that operator must turn off the machine and stop the processing to execute PM. We assume that the pre-emption of operations is not allowed, i.e., the non-resumable case is considered here. In one PM period, the machine's effective working time cannot be larger than a threshold $PT_j$. The machine's age is defined here to explain the PM constraint. The machine's age

equals to zero at the beginning of horizon. It does not change if machine does not process jobs. The age of machine $M_j$ is increased by $p_{ij}^0$ after processing the operation $O_{ij}$. In addition, the machine's age becomes zero immediately after a PM. Therefore, the machine's age cannot exceed $PT_j$ at any time for each machine according to the PM constraint.

There are three states for one machine: off/idle/work. It is obvious that the power demand of machine is zero when it is off. When one machine is running and no operation is being processed on this machine, it is idle. The energy consumption per unit time of $M_j$ equals $e_j^{id}$ when it is idle. It is straightforward that shutting down the machine is a good method to save energy when it is idle. However, when the machine is setup again, it consumes additional energy from the off state to the running state. It is assumed that energy consumption of $M_j$ caused by setup equals $e_j^{st}$, which is a constant. Then, the energy consumption of $M_j$ during idle time and $e_j^{st}$ should be compared when deciding to shut it down. The setup time is very small, which is ignored in this paper. At the start of the production horizon, each machine is off. The machine needs to be turned on to process the first job and turned off after finishing all jobs. The additional on/off operations during the scheduling horizon need to be decided according to the planning of production and PMs.

When a job is being processed in one machine, the speed level of machine needs to be selected for the working state. There is a finite and discrete set of levels $Level_j = \{1, 2, \ldots, L_j\}$ for machine $M_j$. Accordingly, the speed set is $\left\{v_j^1, v_j^2, \ldots, v_j^{L_j}\right\}$ for different levels and the power demand set is $\left\{e_j^1, e_j^2, \ldots, e_j^{L_j}\right\}$. The actual processing time of $O_{ij}$ equals to $p_{ij}^0/v_j^1$ when $M_j$ is in Level 1; meanwhile, the energy consumption of $M_j$ equals $e_j^1$ during one unit time. It is obvious that the actual processing time is shorter with a higher speed level and the power demand is larger at the same time. In addition, we assume that $e_j^1\left(p_{ij}^0/v_j^1\right) < e_j^2\left(p_{ij}^0/v_j^2\right)$, which means the total energy consumption of one operation is larger when the speed level is higher. It is practical in industry and also approved in the literature.

The total energy consumption during the production horizon consists of the machines' setup consumption and the energy consumption during working time and idle time. The production planning needs to be optimized for minimizing the total energy cost. Meanwhile, the peak demand constraint must be obeyed, i.e., the maximum power demand of production system must be smaller than a promised threshold $\overline{D}$. It is common that the power utility proposes this requirement when it signs the contract with its industry customer. The peak demand is defined as the highest average kW measured in each interval of length $\delta$ (usually 15 min) during the production horizon. We assume that $\overline{D}$ is smaller than $\sum_{j=1}^{m} e_j^{L_j}$. Otherwise, all speed combinations are feasible, which means the peak demand constraint is too loose. We assume that $\overline{D}$ is larger than $\sum_{j=1}^{m} e_j^1$. Otherwise, the peak demand constraint is too tight. The manufacturing plant negotiates with the power utility to increase its maximum allowed power demand.

As mentioned above, three interrelated aspects are integrated into our decision model. First, the production sequence and the speed level for each operation need to be determined. Second, the maintenance planning needs to be determined while the machine's age cannot exceed a given threshold. Third, the machines' off/on need to be decided to minimize the total energy cost with the consideration of peak demand constraint. One integrated solution for an example with three machines and five jobs is shown in Figure 1. Different jobs are presented by different colors. Black box means the machine is off during this time. Box with shadow lines means that a PM is performing during this time. The number in "{}" on the box means the speed level of this machine during this time. The length of each interval is $\delta$. The time window of the fourth interval is $[t_3, t_4]$. In this case, all jobs are finished before the end time line.

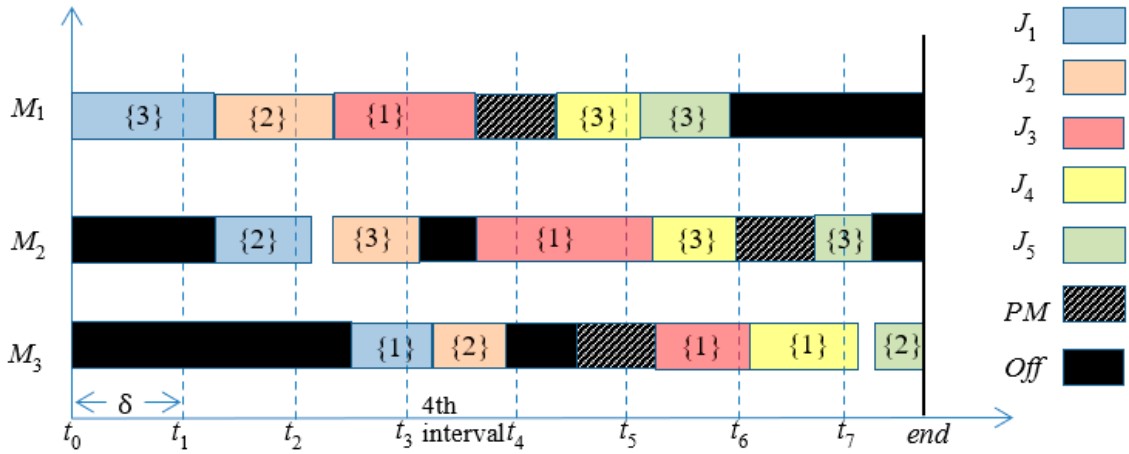

**Figure 1.** Gantt chart of the integrated solution.

In machine $M_1$, a PM is performed between $J_3$ and $J_4$. Otherwise, the machine's age would be larger than the threshold after $J_4$ without this PM. In machine $M_2$, idle time exists between $J_1$ and $J_2$. The machine is not shut down since the idle time is very short, which results in that idle consumption being smaller than $e_2^{st}$. Correspondingly, there is one setup between $J_2$ and $J_3$. In machine $M_3$, job $J_1$ is started later than its finish time in $M_2$ since the average power demand during $[t_2, t_3]$ must be smaller than $\overline{D}$. Furthermore, the time length is very long between $J_2$ and $J_3$ since $J_3$ can only be started after its finish time in $M_2$. Thus, $M_3$ is shutdown to save energy after finishing $J_2$. Since the time length is longer than the maintenance time, it is a good opportunity to perform a PM, even though the machine's age does not reach the threshold.

Now, the average power demand in the 4th interval is calculated here. Let $f_{31}$ be the finish time of $J_3$ in machine $M_1$. Let $f_{22}$ be the finish time of $J_2$ in machine $M_2$. Let $s_{32}$ be the start time of $J_3$ in machine $M_2$. Let $f_{13}$ be the finish time of $J_1$ in machine $M_3$. Let $p_{23}$ be the actual processing time of $J_2$ in machine $M_3$. The total energy consumption can be obtained as follows.

$$TEC = (f_{31} - t_3)e_1^1 + (f_{22} - t_3)e_2^3 + e_2^{st} + (t_4 - s_{32})e_2^1 + (f_{13} - t_3)e_3^1 + p_{23}e_3^2.$$

Thus, the average power demand in the fourth interval equals to $TEC/\delta$, which must be smaller than the given $\overline{D}$.

For the traditional flow shop scheduling problems without the consideration of energy-related objective, non-delay schedule is optimal for minimizing makespan, which is proved to be NP-hard. In our study, the integrated problem is much more complicated than the traditional problem. Considering the jobs' sequence, speed level, PMs, and machine's on/off, the searching space size is $(n!)\left[\prod_{j=1}^{m}(L_j)^n\right](2^{nm})(2^{nm})$. Besides, additional buffer times need to be inserted into the schedule since the peak demand constraint must be obeyed. Thus, the integrated problem is a mixed integer programming problem with discrete variables and continuous variables.

### 3.2. Mathematical Programming Model

Considering the makespan and the total energy consumption, a bi-objective mathematical model is established as follows. Considering the peak demand, the time window of the $t$th interval $[WS_t, WE_t]$ equals $[(t-1)\delta, (t)\delta]$. Furthermore, $T$ is the number of intervals.

Let $O_{[k]j}$ be the $k$th operation on machine $M_j$. Let $\overline{H}$ be a very large constant. Let $\underline{H}$ be a very small constant which is larger than zero.

Decision variables:

$x_{i[k]jl}$ : If job $i$ is processed at the $k$th position on machine $M_j$ with speed $l$, it is 1; else, 0.

$y_{[k]j}$ : If there is a PM immediately before $O_{[k]j}$, $y_{[k]j} = 1$; else, 0.

$z_{[k]j}$ : If there is a setup immediately before $O_{[k]j}$, $z_{[k]j} = 1$; else, 0.

　　Auxiliary decision variables:

$z_{t[k]j}$ : If $O_{[k]j}$ is started in the $t$th interval and there is a setup before $O_{[k]j}$, it is 1; else, 0.

$p_{[k]j}$ : Actual processing time of $O_{[k]j}$.

$d_{[k]j}$ : Actual energy consumption per unit time when processing $O_{[k]j}$.

$s_{[k]j}$ : Start time of $O_{[k]j}$.

$c_{[k]j}$ : Finish time of $O_{[k]j}$.

$b_{[k]j}$ : Machine's age immediately before $O_{[k]j}$.

$a_{[k]j}$ : Machine's age immediately after $O_{[k]j}$.

$E_{tj}$ : Energy consumption of machine $M_j$ during the $t$th interval.

$E^1_{t[k]j}$ : Energy consumption of $M_j$ caused by $O_{[k]j}$ during the $t$th interval.

$E^2_{t[k]j}$ : Energy consumption of $M_j$ caused by idle time before $O_{[k]j}$ during the $t$th interval.

　　Objectives:

$$\text{Min } C_{max} = c_{[n]\text{m}} \tag{1}$$

$$\text{Min } TEC = \sum_t^T \sum_{j=1}^m E_{tj} \tag{2}$$

　　Constraints:

$$\sum_{k=1}^n \sum_{l=1}^{L_j} x_{i[k]jl} = 1 \forall i; \forall j \tag{3}$$

$$\sum_{i=1}^n \sum_{l=1}^{L_j} x_{i[k]jl} = 1 \forall k; \forall j \tag{4}$$

$$\sum_{l=1}^{L_j} x_{i[k]jl} = \sum_{l=1}^{L_j} x_{i[k]hl} \forall j, h \in M; \forall i; \forall k \tag{5}$$

$$p_{[k]j} = \sum_{i=1}^n \sum_{l=1}^{L_j} x_{i[k]jl}\, p_{ij}^0 / v_j^l \;\; \forall k; \forall j \tag{6}$$

$$d_{[k]j} = \sum_{i=1}^n \sum_{l=1}^{L_j} x_{i[k]jl} e_j^l \;\; \forall k; \forall j \tag{7}$$

$$c_{[k]j} = s_{[k]j} + p_{[k]j} \;\; \forall k; \forall j \tag{8}$$

$$s_{[k]j} \geq c_{[k]j-1} \;\; \forall k; \forall j \tag{9}$$

$$s_{[k]j} \geq c_{[k-1]j} + pt_j y_{[k]j} \forall k; \forall j \tag{10}$$

$$a_{[k]j} = b_{[k]j} + \sum_{i=1}^n \sum_{l=1}^{L_j} x_{i[k]jl}\, p_{ij}^0 \;\; \forall k; \forall j \tag{11}$$

$$b_{[k+1]j} = a_{[k]j}\big(1 - y_{[k+1]j}\big) \forall k; \forall j \tag{12}$$

$$a_{[k]j} \leq PT_j \;\; \forall k; \forall j \tag{13}$$

$$y_{[k]j} \leq z_{[k]j} \forall k; \forall j \tag{14}$$

$$z_{[1]j} = 1 \;\; \forall j \tag{15}$$

$$\sum_{t=1}^T z_{t[k]j} = z_{[k]j} \forall k; \forall j \tag{16}$$

$$WS_t z_{t[k]j} \leq s_{[k]j} \leq [WE_t - \underline{H}] z_{t[k]j} + \big(1 - z_{t[k]j}\big)\overline{H} \;\; \forall t; \forall k; \forall j \tag{17}$$

$$E_{tj} = \sum_{k=1}^n \Big(E^1_{t[k]j} + E^2_{t[k]j} + z_{t[k]j} e_j^{st}\Big)\forall t; \forall j \tag{18}$$

$$E^1_{t[k]j} = \big\{\max\big(0, \min\big(c_{[k]j}, WE_t\big) - \max\big(s_{[k]j}, WS_t\big)\big)\big\}d_{[k]j} \;\; \forall t; \forall k; \forall j \tag{19}$$

$$E^2_{t[k]j} = \left\{\max\left(0, \min\left(s_{[k]j}, WE_t\right) - \max\left(c_{[k-1]j}, WS_t\right)\right)\right\} e^{id}_j \left(1 - z_{[k]j}\right) \tag{20}$$

$$\sum_{j=1}^m E_{tj} \leq \overline{D} \forall t \tag{21}$$

$$x_{i[k]jl}, \ y_{[k]j}, \ z_{[k]j}, \ z_{t[k]j} \text{ are binaries; others are continuous variables} \tag{22}$$

The objectives in Equations (1) and (2) are two kinds of considerations, which show the tradeoff between two conflict aspects. The constraint in Equation (3) ensures that one job must be located in one position of each machine and can only be processed with one speed. The constraint in Equation (4) ensures that one position of each machine can only be engaged by one job. The constraint in Equation (5) ensures that the jobs' sequences on different machines are the same. The constraints in Equations (6) and (7) specify the actual processing time and power demand of the operation $O_{[k]j}$. The constraint in Equation (8) shows the relation between the start and the finish of $O_{[k]j}$. The constraint in Equation (9) ensures that one job can only be started after it is finished on the last machine. The constraint in Equation (10) ensures that one machine can only process one task at any time. The constraints in Equations (11) and (12) derive the machines' age at different time. The constraint in Equation (13) is the PM period constraint. The constraint in Equation (14) means that one PM can only be processed when machine is off. The constraint in Equation (15) means that each machine needs to be turn on to start processing jobs at the beginning. The constraints in Equations (16) and (17) mean that the setup can only be located in one interval if there is one setup before one operation. The constraint in Equation (18) means that the energy consumption of machine $M_j$ during $t$th interval is caused by three parts: processing jobs, idle time, and setup. The constraint in Equation (19) specifies the overlap between the processing time of each job on $M_j$ and the $t$th interval, based on which the energy consumption can be obtained using the power demand per unit time. The constraint in Equation (20) shows that, if one setup exists in the idle time between $O_{[k-1]j}$ and $O_{[k]j}$, then no energy consumption occurs in this idle time; otherwise, the idle-time energy consumption needs to be accounted. The constraint in Equation (21) is the peak demand constraint, which requires the average energy consumption in each interval below the threshold. The constraint in Equation (22) shows the features of decision variables.

## 4. Algorithm Designing

The mathematical model cannot be solved effectively by the commercial software, such as Cplex, Grobi, and Lingo. One reason is that the constraints in Equations (12), (19), and (20) are nonlinear. Although it is possible to get the linear formulas using the big constant method and additional 0/1 variables, too many constraints and variables would be added into the model. Besides, the flow shop scheduling problem is still very complicated even if the energy-related factors are ignored. Thus, we adopt the meta-heuristic to solve this model instead of using exact algorithms. According to the research results in [55], meta-heuristics such as genetic algorithm (GA) [56], ant colony optimization algorithm (ACO) [57], and particle swarm optimization algorithm (PSO) [58] are very effective to solve the combinatorial problems such as flow shop, job shop and open shop scheduling problems to get the near optimal solutions.

For the multi-objective optimization problem minimizing a vector of objective functions, the model can be simply formulated as Min $Obj(\pi) = \{Obj_1(\pi), Obj_2(\pi), \ldots, Obj_\tau(\pi)\}$ subjected to constraints. To define the optimal solution for the multi-objective optimization problem, we introduce the concept of Pareto firstly. Let $\pi_1$ and $\pi_2$ be two solutions for the given problem. If $\pi_1$ is worse than $\pi_2$ from the viewpoints of all objectives, then $\pi_1$ is dominated by $\pi_2$, i.e., $\pi_2 \prec \pi_1$. If there is no solution $\pi$ satisfying $\pi \prec \pi_2$, then solution $\pi_2$ is Pareto optimal. The set of all Pareto optimal solutions is denoted by the Pareto frontier.

For the single-objective problem, the aim of model-solving is to find the best solution with minimum objective value. For the multi-objective problem, the aim of model-solving is to find the Pareto frontier. For any two solutions $\pi_1$ and $\pi_2$ on the Pareto frontier, there is no dominance relation between $\pi_1$ and $\pi_2$, i.e., $\pi_1$ is better than $\pi_2$ for some objectives and $\pi_1$ is worse than $\pi_2$ for the other

objectives at the same time. Some researchers design a weighted sum of all objectives and use the single-objective algorithm to solve the model, which can only get one solution each time for the defined weights. Instead, we adopt the multi-objective optimization algorithm based on non-dominated sorting to search the Pareto frontier directly.

## 4.1. Framework

Deb et al. [59] designed a multi-objective evolutionary algorithm based on the non-dominated sorting strategy and crowding distance, which is denoted by *NSGA-II*. The performance of an individual in the population is evaluated by the rank and crowding distance. The basic framework of *NSGA-II* is similar to the traditional GA solving single objective problem. The main difference is how to get new population in the evolution. In *NSGA-II*, the parent population and the offsprings are merged into one pool. Then, the best individuals are selected from the pool to compose the new population. The detailed structure can be found in [59].

Here, we only provide the basic framework of *NSGA-II* in order to make the methodology clear and ignore the detailed sorting procedures calculating the rank and crowding distance. As shown in Figure 2, the population evolves from old generation to new generation according to the survival of the fittest. Since the individual with better fitness has a larger probability to generate offspring, the characteristic of this individual will be likely inherited by the offsprings. Thus, the performance of generation $g+1$ will be better than that of generation $g$. After evolution, the Pareto optimal solutions are obtained from the last generation $g_{max}$.

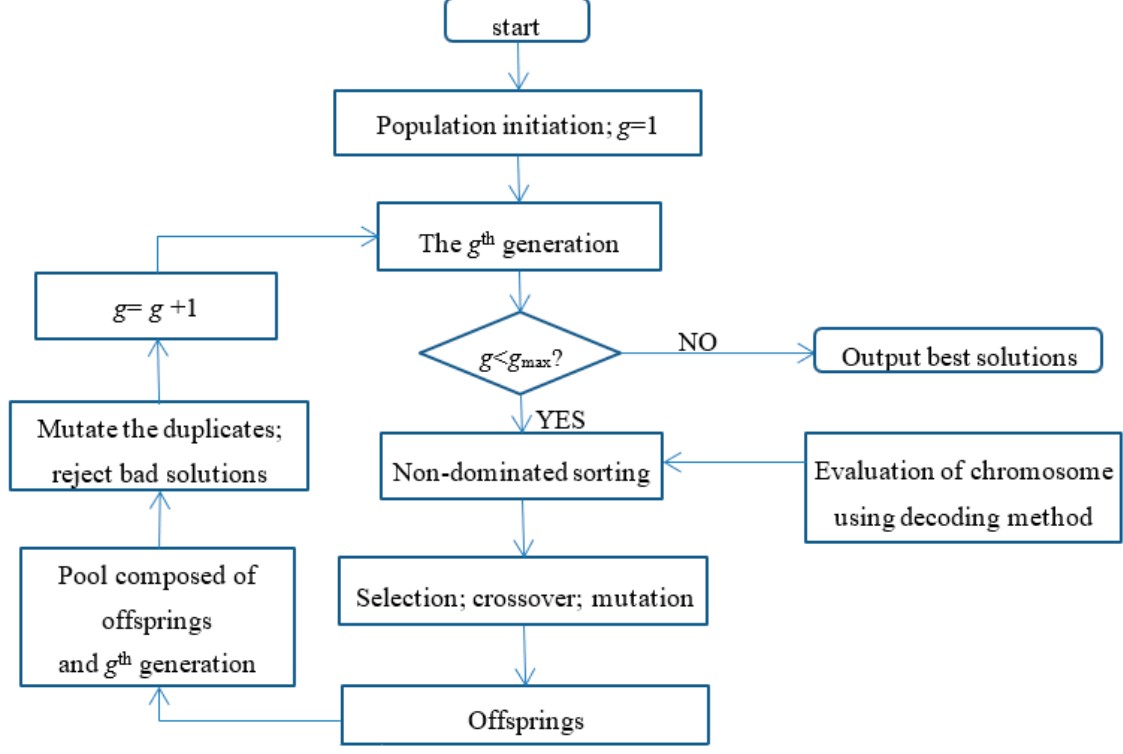

**Figure 2.** The flowchart of *NSGA-II*.

We also provide the particular genetic operators of our algorithm in order that the readers can repeat our study.

The chromosome representation. We use two parts to represent a chromosome. Part one is a string of integers, which shows the sequence of processed jobs. For example, {3, 4, 2, 5, 1} implies that job $j_3$ is processed in the first position. Part two is the speed level selected for each operation on each machine, which is a matrix. For example, when there are three machines, {1, 2, 2, 1, 3; 2, 1, 2, 4, 1; 5, 3, 1,

5, 5} implies that the first processed operation in machine $M_1$ is with speed Level 1, the first operation in machine $M_2$ is with speed Level 2, and the first operation in machine $M_3$ is with speed Level 5.

The chromosome evaluation. Based on the chromosome, the main planning is already there. We can get the actual processing time and energy consumption for each operation. Then, we need to optimize $y$ and $z$ minimizing the objectives without violating the constraints. The detailed procedure can be found in Section 4.2. After finding the optimal $y$ and $z$, the makespan and energy consumption can be obtained for the chromosome. If $C_{max}(ch_1) \leq C_{max}(ch_2)$ and $TEC(ch_1) \leq TEC(ch_2)$, then $ch_2$ is dominated by $ch_1$, i.e., $ch_1$ has a better rank. Otherwise, two chromosomes are located on the same frontier, i.e., they have the same rank. For the chromosomes on the same frontier, the Euclidian distance between chromosomes needs to be calculated. $ch_1$ is better than $ch_2$ if they are on the same frontier and the obtained crowding distance based on Euclidian distance of $ch_1$ is larger.

The selection. Similar to traditional GA, the fitness value based on the performance of chromosome needs to be calculated. In this paper, the fitness value of chromosome is a combination of rank and crowding distance. The chromosome with a better fitness value is a parent with a higher probability compared with the worse ones. To reduce the selection pressure and increase the diversity of population, we use the two-size tournament method. We choose two chromosomes randomly and put the better one into the mating pool. Furthermore, all the other parents are chosen using the same method.

The crossover. We choose two parents $ch_1$ and $ch_2$ from the mating pool and perform the crossover procedure with a probability of 0.8. For the first part of chromosome, we use the order crossover method. A string of $n$ binaries is generated stochastically. For example, a string of {1, 0, 1, 0, 0} is generated when $n = 5$. Firstly, the offspring's gene corresponding to the "1" of the string is copied from the parent $ch_1$. Secondly, these integers are deleted from the parent $ch_2$. Thirdly, the offspring's gene corresponding to the "0" of the string is copied from the parent $ch_2$ in sequential. For the second part of chromosome, partial crossover is adopted. For the speed level of each machine, the front of offspring is the same with the front of parent $ch_1$ and the rear of offspring is the same with the rear of parent $ch_2$.

The mutation. We choose one offspring $ch$ and perform the mutation procedure with a probability of 0.05. For the first part of chromosome, we use the inversion mutation method. Firstly, two genes $g_1$ and $g_2$ are selected from the string randomly. Secondly, we inverse the numbers in the range of $[g_1, g_2]$ for the chromosome. For the second part of chromosome, one gene is randomly chosen. Then, the operator changes the corresponding speed level randomly.

The population management. The population is composed of 200 individuals. In the initial population, 198 individuals are randomly created and two solutions are constructed heuristically as follows. The first solution corresponds to the low-speed mode, which means the speed levels of all machines are set to the lowest level for processing all jobs. The second solution corresponds to the high-speed mode, which means the speed levels of all machines are set to the highest level for processing all jobs. After selection, crossover, and mutation, we get 200 offsprings. Then, 200 offsprings and 200 individuals from the old population are added into one single pool. To keep good population diversity, we check the pool to find the duplicates and use the mutation method to change them. Finally, we select the 200 best individuals from the pool to compose the new population. The evolution stops when the number of iterations $g$ reaches the maximum limit $g_{max}$.

## 4.2. Decoding Method

For a chromosome $ch$, the jobs' sequence and speed level are fixed. We only need to find the optimal $y$ and $z$ to evaluate the performance of $ch$. Since the PM and machine's on/off can only be executed during idle time between two consecutive jobs, analysis about idle time needs to be done.

Considering the objective of total energy consumption, machine should be turn off when the energy consumption during idle time is larger than setup consumption. Meanwhile, PM should be performed here if the length of idle time is longer than $pt_j$; otherwise, PM is ignored here unless the maintenance period constraint will be violated without a PM.

When the energy consumption during idle time is smaller than setup consumption, we should keep machine on for saving total consumption. However, if a PM has to be performed here considering the maintenance period constraint, we must turn off the machine since $y_{[k]j} \le z_{[k]j}$.

Using the rules shown in above two paragraphs, we can get the start times of all operations from beginning to the end on each machine if this schedule does not violate the peak demand constraint. Then, the method dealing with peak demand constraint is described here. To guarantee the energy consumption in each interval is below the peak demand threshold, we schedule the operations from the first interval to the last interval one by one. The time window of the $t$th interval is $[WS_t, WE_t]$. Let $J_{tj}$ be the job started before $WS_t$ and still not finished yet at $WS_t$ on machine $M_j$. $p^0_{J_{tj}}$ is the basic processing time of this job. $v^l_{J_{tj}}$ is speed level obtained from the chromosome. Then, we can get the finish time of $J_{tj}$, which is $c_{J_{tj}}$. Considering the demand level $d^l_{J_{tj}}$, we can get the energy consumption $D^A_t$ of all started jobs during the $t$th interval, which equals $\sum_{j=1}^{m}\left\{\min\left(WE_t, c_{J_{tj}}\right) - WS_t d^l_{J_{tj}}\right\}$. If $D^A_t > \overline{D}$, the peak demand constraint has already been violated. Then, we need to modify the speed level of $J_{tj}$ in the chromosome to reduce $D^A_t$. When $D^A_t \le \overline{D}$, we can always find a feasible schedule for the chromosome by delaying the follow-up operations, which is shown in the following procedure.

Let $age_j$ be the age of machine $M_j$. Let $off_j$ be the state of machine $M_j$. The equation $off_j = 0$ means that $M_j$ is shutdown. Otherwise, $off_j = 1$ and $M_j$ is running.

Step 1.　Set $age_j = 0; off_j = 0 \forall j$. Set $t = 1$.

Step 2.　Set $D^A_t = 0$.

Step 3.　Calculate the energy consumption of $J_{tj} \forall j$ in the $t$th interval, which is $D\left(J_{tj}\right)$. Then, we have $D^A_t = \sum_{j=1}^{m} D\left(J_{tj}\right)$. If $D^A_t > \overline{D}$, then change the speed level of $J_{tj}$ and update $D^A_t$. Calculate the finish time of $J_{tj}$ for each machine, which is $f\left(J_{tj}\right)$.

Step 4.　Set $j = 1$.

Step 5.　If $j > m$, then set $t = t + 1$, go to Step 2. Else, if $f\left(J_{tj}\right) > WE_t$, set $j = j + 1$, go to Step 6. Else, go to Step 6.

Step 6.　If $D^A_t \ge \overline{D}$, then set $off_j = 0$, $j = j + 1$, go to Step 5; else, go to Step 7.

Step 7.　Let $J_{tjs}$ be the successor of $J_{tj}$. Let $p_0\left(J_{tjs}\right)$ be the basic processing time of $J_{tjs}$. If $p_0\left(J_{tjs}\right) + age_j > PT_j$, then set $off_j = 0$ and perform a PM to set $age_j = 0$.

Step 8.　Find the earliest allowed start time of $J_{tjs}$, which is $s\left(J_{tjs}\right)$. If $s\left(J_{tjs}\right) > WE_t$, then go to Step 9; else, go to Step 11.

Step 9.　If $off_j = 0$, then set $j = j + 1$, go to Step 5; else, go to Step 10.

Step 10.　Calculate the energy consumption of the remaining idle time of this interval, which is $Idle_t$. If $Idle_t + D^A_t \le \overline{D}$, then set $D^A_t = D^A_t + Idle_t$, $j = j + 1$, go to Step 5. Else, set $off_j = 0$, $j = j + 1$, go to Step 5.

Step 11.　If $off_j = 1$, then try to start the job $J_{tjs}$ as early as possible, update $D^A_t$, go to Step 12; else, go to Step 14.

Step 12.　If the idle consumption between $J_{tjs}$ and $J_{tj}$ is smaller than $e^{st}_j$, then set $off_j = 0$. If the length of this idle time is longer than $pt_j$ and $off_j = 0$, then set $age_j = 0$. Go to Step 13.

Step 13.　If the finish time of $J_{tjs}$ is smaller than $WE_t$, then set $age_j = age_j + p_0\left(J_{tjs}\right)$, consider the next operation in this machine, go to Step 7. Else, this job will be one job started in the $t$th interval and finished in the following intervals, set $j = j + 1$, go to Step 5.

Step 14.　If $e^{st}_j + D^A_t < \overline{D}$, then set $off_j = 1$, $D^A_t = D^A_t + e^{st}_j$, go to Step 11. Else, set $j = j + 1$, go to Step 5.

## 5. Numerical Results

The algorithm was programmed in the C# platform and the program was run on a Hewlett-Packard laptop with an Intel Core i5 2.50 GHz CPU and 8 GB RAM.

### 5.1. Validation of Algorithm

The problem parameters are set as follows. The length of interval, $\delta$, equals 15 min. The basic processing times of operations are uniformly generated from an interval [10 min, 60 min]. The length of PM period equals 24 h. For each machine, there are six speed levels. Accordingly, the speed set is {1, 1/0.9, 1/0.8, 1/0.7, 1/0.6, 2} for different levels and the power demand set is {1, 1.2, 1.5, 1.8, 2.2, 2.8}. Then, the energy consumption per unit time (1 min) equals 1 unit if the machine processes a job with speed Level 1. The energy consumption per unit time is 0.5 units when the machine is idle. The energy consumption caused by one setup equals 10 units. The production line is composed of five machines. The number of jobs $n$ is set to {50, 100, 200}. The length of maintenance is set to {60 min, 180 min, 300 min}. The peak demand threshold $\overline{D}$ is set to {8, 9, 10, 11, 12}. Accordingly, the energy consumption limit in one interval equals to {120, 135, 150, 165, 180}. Thus, there are $3 \times 3 \times 5 = 45$ scenarios. For each scenario, three instances are generated randomly, which leads to 135 instances in total.

GA is a kind of algorithm that less depends on the parameters. There are only four parameters in this algorithm: population size, crossover probability, mutation probability, and the maximum generation. For the population size, a larger size usually brings a larger potential to find good solutions. However, a larger size means more computation time. Considering the number of jobs, the size is set to 200. Usually, a large crossover probability is recommended to increase the search ability of GA; a small mutation probability is recommended to guarantee the stability of GA. After calibration on a small set of random instances to evaluate the performance of different probabilities, the crossover probability is set to 0.8 and the mutation probability is set to 0.05. For the maximum generation, it is obvious that better solutions can be found with the evolution of population and more computation time is required for a larger maximum limit. Here, we show the details of one instance in Figure 3. In the 50th generation, 99 solutions are obtained on the frontier. In the 100th generation, 156 solutions are obtained on the frontier. In the 1000th generation, 200 solutions are obtained on the frontier, which means all solutions of population are located on the frontier. The frontier obtained in the 3000th generation is very close to that in 1000th generation, which shows the convergence of our algorithm. Thus, we set the maximum generation 3000 in the following numerical experiments.

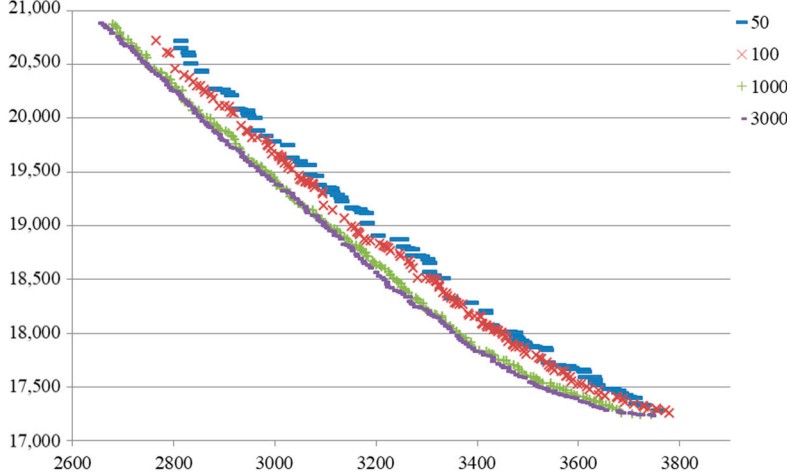

**Figure 3.** The frontiers obtained in different generations.

Since it is the first attempt to solve this integrated problem, no algorithm can be found in the literature to solve the model proposed in this paper. Thus, we need to design a method as the benchmark to validate the effectiveness of *NSGA-II*. It is well known that *NEH* heuristic is very effective when minimizing the makespan of flow shop scheduling problem without the energy-related consideration. It is also common that the decisions of production and maintenance are determined independently by

two departments in sequential. Combining the idea of *NEH* and the realistic situation in industrial plants, a constructive heuristic *CH* is designed as follows.

First, one machine can only process all jobs in a same speed level. Since there are five machines and six levels, we can get $6^5 = 7776$ combinations from {1,1,1,1,1} to {6,6,6,6,6}. Second, for each speed combination, we can get the actual processing times of all jobs in all machines. Using the *NEH* method, we can get a sequence of jobs for this speed combination. Then, ignoring the peak demand constraint, the PMs are inserted as late as possible into the production planning and the machines' on/off is determined aiming at reducing the energy consumption. It means that we construct 7776 solutions. Finally, we select and keep the solution if it satisfies the following two conditions: (1) the solution does not violate the peak demand constraint at any time; and (2) the solution is not dominated by any other feasible solutions obtained by *CH*.

For each instance, we compare the solutions obtained by *NSGA-II* and *CH*. It is not easy to judge the comparison between two multi-objective algorithms since a set of Pareto solutions are obtained using the algorithm instead of one optimal solution. Coverage metric is the most famous criteria to evaluate the difference between two Pareto frontiers. Let *A* denote NSGA-II. Let *B* denote *CH*. Then, the dominance relationship between the solutions in two frontiers can be evaluated by the value of $C(A, B)$, which is calculated by $C(A, B) = \frac{|\{b \in B | \exists a \in A : a \geq b\}|}{|B|}$. $|B|$ is the number of solutions on the frontier of *B*. $a \geq b$ means that solution *b* is dominated by solution *a*. If $C(A, B) = 1$, then all solutions obtained by *B* are dominated by at least one solution obtained by *A*.

The comparison results are shown in Table 2.

**Table 2.** The coverage metric between *NSGA-II* and *CH*.

| *pt* | *n* | Peak Limit | | | | | | | | | |
|---|---|---|---|---|---|---|---|---|---|---|---|
| | | 120 | | 135 | | 150 | | 165 | | 180 | |
| | | *C(A,B)* | *C(B,A)* | *C(A,B)* | *C(B,A)* | *C(A,B)* | *C(B,A)* | *C(A,B)* | *C(B,A)* | *C(A,B)* | *C(B,A)* |
| | 50 | 1 | 0 | 1 | 0 | 0.96 | 0 | 1 | 0 | 1 | 0 |
| 60 | 100 | 1 | 0 | 1 | 0 | 0.94 | 0 | 0.87 | 0 | 1 | 0 |
| | 200 | 0.98 | 0 | 0.98 | 0 | 0.89 | 0 | 0.77 | 0 | 0.94 | 0 |
| | 50 | 1 | 0 | 1 | 0 | 1 | 0 | 1 | 0 | 1 | 0 |
| 180 | 100 | 1 | 0 | 1 | 0 | 0.98 | 0 | 1 | 0 | 1 | 0 |
| | 200 | 1 | 0 | 1 | 0 | 0.97 | 0 | 1 | 0 | 1 | 0 |
| | 50 | 1 | 0 | 1 | 0 | 1 | 0 | 1 | 0 | 0.99 | 0 |
| 300 | 100 | 1 | 0 | 1 | 0 | 1 | 0 | 1 | 0 | 1 | 0 |
| | 200 | 1 | 0 | 1 | 0 | 1 | 0 | 1 | 0 | 1 | 0 |

Table 2 shows that most of solutions obtained by *CH* are dominated by *NSGA-II*. Furthermore, no solutions obtained by *NSGA-II* are dominated by *CH* in all instances. Since Table 2 only shows the performance in percentage, we also record the number of solutions obtained by algorithms in Table 3. In this table, we can find that the number of solutions obtained by *NSGA-II* is much larger than that of *CH*. For the *NSGA-II*, the population size is 200, which means that all individuals in the population after evolution are located on the first frontier. For the *CH*, the number of solutions is larger for the instances with smaller *n* and larger peak limit. The reason behind this fact is that more solutions can be kept at the last step of *CH* when peak demand constraint is looser. However, the performance of *CH* is very poor considering that 7776 solutions are generated at the beginning of *CH*. More solutions will provide more choices for the managers.

The metric can only tell us which frontier is better. However, the extent of improvement and the solution diversity cannot be obtained via the values in the metric. Thus, we draw the Pareto frontiers obtained by different algorithms in Figure 4 for the instance when peak limit is 150 and maintenance time is 300. As shown in Figure 4a–c, *NSGA-II* performs much better than *CH* in solution quality and diversity. It also provides us an intuitive illustration for the conclusions obtained from the metric. Meanwhile, we record a "FAST" solution in these figures. A "FAST" solution is that all machines process jobs at the highest speed level to finish all jobs as quickly as possible. Although the makespan

of "FAST" solution is about 10% shorter than the fastest solution obtained by *NSGA-II*, we find that the peak demand of system is larger than the threshold during more than half intervals for the "FAST" solution. It means "FAST" solution is strongly infeasible.

**Table 3.** The number of solutions in the Pareto frontiers obtained by *NSGA-II* and *CH*.

| pt | n | Peak Limit | | | | | | | | | |
|----|---|------|----|------|----|------|----|------|----|------|----|
| | | 120 | | 135 | | 150 | | 165 | | 180 | |
| | | *NS-II* | *CH* | *NS-II* | *CH* | *NS-II* | *CH* | *NS-II* | *CH* | *NS-II* | *CH* |
| | 50 | 200 | 18 | 200 | 25 | 200 | 30 | 200 | 35 | 200 | 37 |
| 60 | 100 | 200 | 16 | 200 | 22 | 200 | 27 | 200 | 31 | 200 | 33 |
| | 200 | 200 | 16 | 200 | 21 | 200 | 25 | 200 | 29 | 200 | 32 |
| | 50 | 200 | 17 | 200 | 23 | 200 | 28 | 200 | 31 | 200 | 33 |
| 180 | 100 | 200 | 18 | 200 | 24 | 200 | 28 | 200 | 32 | 200 | 35 |
| | 200 | 200 | 18 | 200 | 24 | 200 | 26 | 200 | 30 | 200 | 33 |
| | 50 | 200 | 19 | 200 | 23 | 200 | 29 | 200 | 33 | 200 | 34 |
| 300 | 100 | 200 | 17 | 200 | 22 | 200 | 26 | 200 | 28 | 200 | 30 |
| | 200 | 200 | 15 | 200 | 21 | 200 | 24 | 200 | 26 | 200 | 29 |

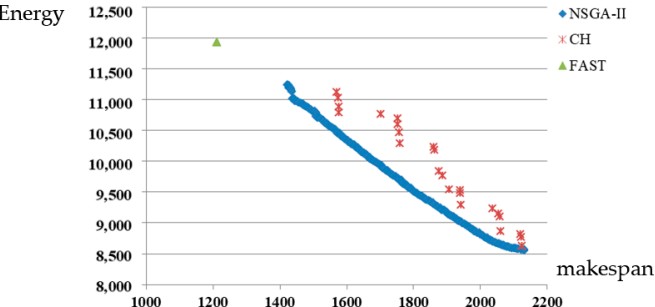

(a) The number of jobs equals 50 (*n*=50)

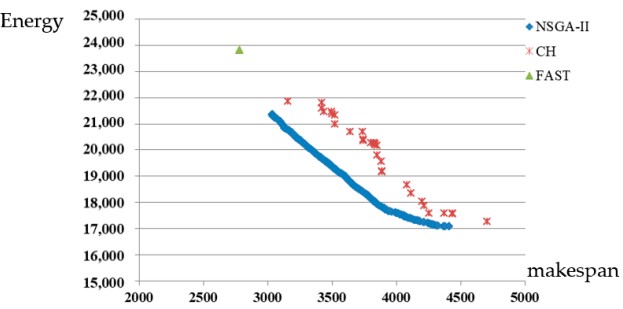

(b) The number of jobs equals 100 (*n*=100)

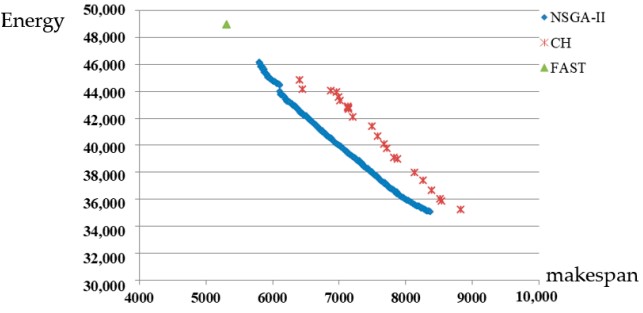

(c) The number of jobs equals 150 (n=200)

**Figure 4.** The comparison between two frontiers under different number of jobs (**a**) The number of jobs equals 50 (n = 50); (**b**) The number of jobs equals 100 (n = 100); (**c**) The number of jobs equals 150 (n = 200).

The computation time of algorithm mainly depends on the number of jobs. The average computation times of *NSGA-II* are 2 ($n = 50$), 8 ($n = 100$), and 12 min ($n = 200$). The average computation times of *CH* are 2 ($n = 50$), 15 ($n = 100$), and 45 min ($n = 200$). Compared with *NSGA-II*, the computation time of *CH* increases rapidly with the increment of $n$. Thus, *NSGA-II* is better than *CH* when solving large-sized problems from the viewpoints of solution accuracy and computation time.

### 5.2. Impact of Constraints

The tradeoff between makespan and energy consumption can be found clearly in Figure 4. In the middle part of *NSGA-II* line, the reduction rate of energy consumption is nearly proportional to the increase rate of makespan. The managers of manufacturing plant need to select appropriate solution based on the production due date. Since PM constraint and peak demand constraint are two important factors in this problem, we show their impacts on the results in this subsection.

Figure 5 shows the difference between solutions obtained by *NSGA-II* under different peak limits. When the peak limit is larger, the length of the line is longer in this figure, which means there are more choices for the managers in this situation. Correspondingly, the managers cannot finish the jobs too early when the peak limit is small even if they can afford a large amount of energy consumption. Besides, the right parts of three lines are almost overlapped. The line under the larger peak limit is slightly better than the line under the smaller peak limit. Then, the impact of peak limit is very small for those managers who want to save the energy consumption with the sacrifice of makespan. On the contrary, the impact of peak limit is large for other managers.

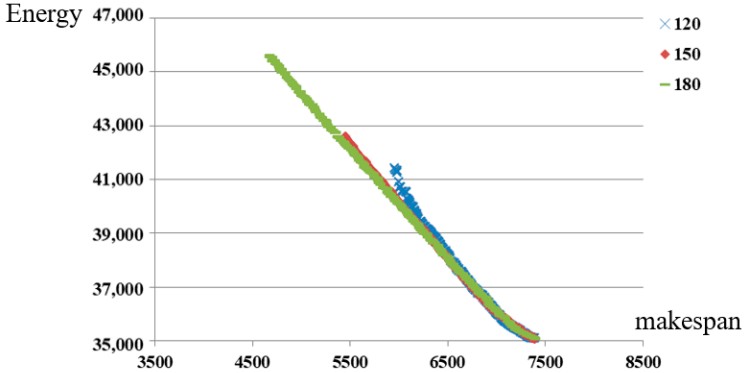

**Figure 5.** The comparison between solutions under different peak limits.

Figure 5 only shows the condition in one random instance. To discover the pattern in all instances, the numerical results are recorded in Table 4. The "left solution" is the extreme solution located at the left end of line, which denotes the solution with the smallest makespan. The "right solution" is the extreme solution located at the right end of line, which denotes the solution with smallest energy consumption. Comparing the solutions under different peak limits, the pattern is exactly the same with that in Figure 5. Besides, the makespan under $n = 100$ is about two times the makespan under $n = 50$. The same trend holds for the energy consumption.

Figure 6 shows the difference between solutions obtained by *NSGA-II* under different maintenances. It is obvious that the solutions are better when the maintenance time is shorter. The minimum energy consumptions are the same in different situations, which means there is the same ultimate limit for the reduction of energy consumption. Meanwhile, the difference between the lengths of three lines is very small. The above pattern not only holds for this random instance, but also holds for all instances. It can be proven by the numerical results in Table 4.

**Table 4.** The numerical results about two extreme solutions.

| pt | n | "Left Solution" | | | | | | "Right Solution" | | | | | |
|---|---|---|---|---|---|---|---|---|---|---|---|---|---|
| | | Peak = 120 | | Peak = 150 | | Peak = 180 | | Peak = 120 | | Peak = 150 | | Peak = 180 | |
| | | $C_{max}$ | TEC | $C_{max}$ | TEC | $C_{max}$ | TEC | $C_{max}$ | TEC | $C_{max}$ | TEC | $C_{max}$ | TEC |
| | 50 | 1438 | 10,117 | 1335 | 10,428 | 1127 | 11,355 | 1896 | 8543 | 1903 | 8539 | 1927 | 8531 |
| 60 | 100 | 2894 | 20,093 | 2653 | 20,878 | 2208 | 23,036 | 3715 | 17,209 | 3744 | 17,233 | 3708 | 17,230 |
| | 200 | 5958 | 41,418 | 5451 | 42,558 | 4674 | 45,587 | 7387 | 35,092 | 7395 | 35081 | 7396 | 35,072 |
| | 50 | 1560 | 10,125 | 1333 | 11,102 | 1211 | 11,489 | 2023 | 8545 | 2023 | 8533 | 2023 | 8551 |
| 180 | 100 | 3101 | 20,271 | 2896 | 21,022 | 2456 | 22,565 | 4021 | 17,112 | 4030 | 17,146 | 4045 | 17,160 |
| | 200 | 6417 | 40,994 | 5965 | 42,450 | 4871 | 47,640 | 7880 | 35,035 | 7846 | 35,049 | 7852 | 35,060 |
| | 50 | 1685 | 10,115 | 1422 | 11,246 | 1337 | 11,421 | 2146 | 8536 | 2132 | 8554 | 2139 | 8557 |
| 300 | 100 | 3364 | 20,011 | 3032 | 21,367 | 2691 | 22,836 | 4315 | 17,101 | 4411 | 17,075 | 4294 | 17,121 |
| | 200 | 6924 | 40,962 | 5813 | 46,168 | 5288 | 48,373 | 8382 | 35,013 | 8378 | 35,053 | 8448 | 35,035 |

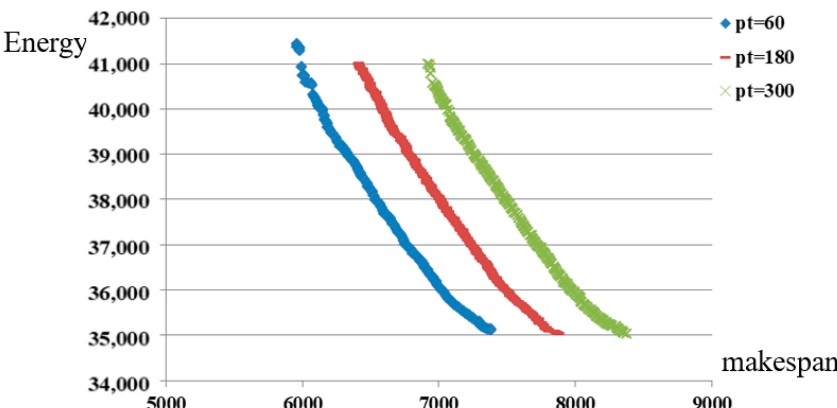

**Figure 6.** The comparison between solutions under different maintenances.

### 5.3. Comparison between Different Solutions

In Table 4, the makespan of the "left solution" is 3101 when *pt* = 180, *n* = 100, and Peak = 120. The energy consumption is 20,271 for this solution. The two objectives of the "right solution" are (4030, 17,146). The detailed energy consumption profiles of these solutions are provided in Figure 7. In the figures, we can find that most of consumptions are caused by processing jobs no matter which solution is considered. Comparing Figure 7a,b, we find that the difference of energy consumption between intervals is rather large for the "left solution". However, the energy consumption equals 75 in many intervals for the "right solution". The reason behind the fact is that each machine processes jobs with speed Level 1 most of the time. Then, five machines consume 5 units per unit time, which results in 5 × 15 = 75 units during one interval.

Next, two other solutions are compared. One is the "FAST" mode solution, which is infeasible. The other one is a feasible solution obtained by modifying the "FAST" solution. The detailed energy consumption profiles of these two solutions are provided in Figure 8. In Figure 8a, we find that the makespan can be very small if we ignore the peak demand constraint. However, when we use this plan in the circumstance with peak demand constraint, its real performance is very bad since Figure 8b shows that the feasible solution has longer makespan and larger energy consumption at the same time.

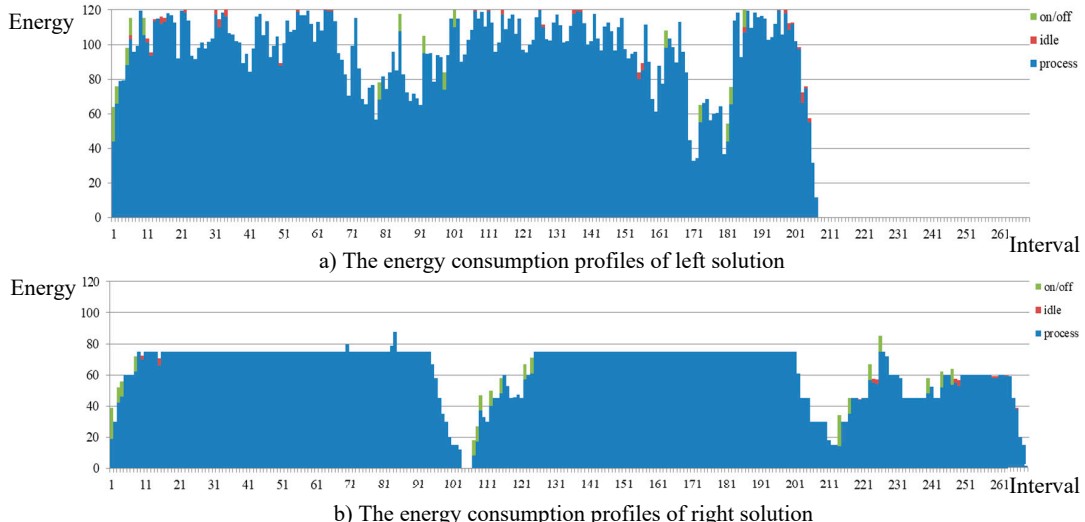

**Figure 7.** The energy consumption profiles of two extreme solutions (**a**) The energy consumption profiles of left solution; (**b**) The energy consumption profiles of right solution.

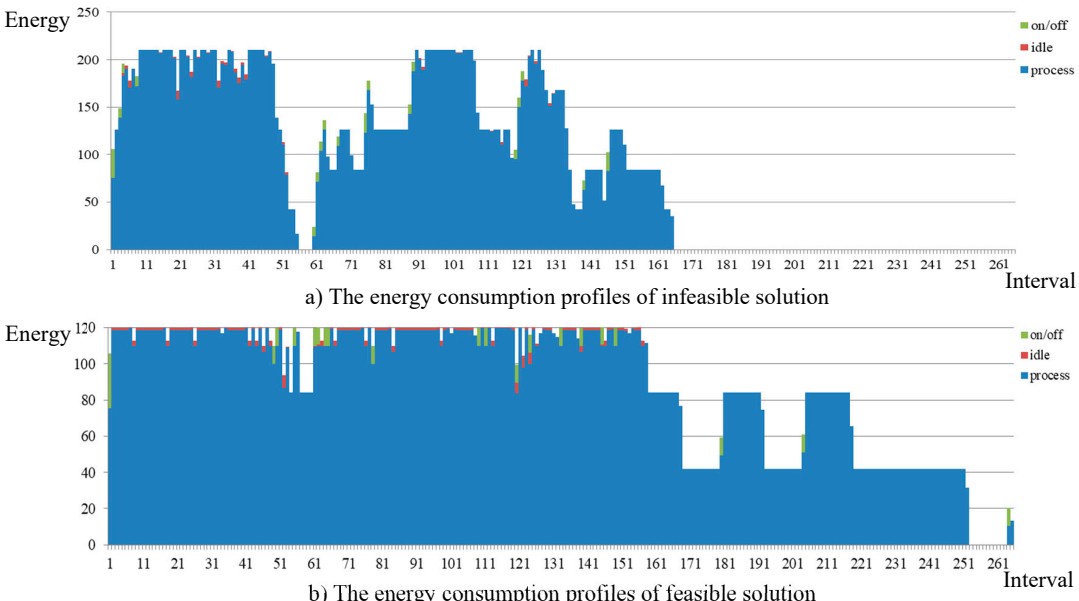

**Figure 8.** The energy consumption profiles of two "FAST" solutions; (**a**) The energy consumption profiles of infeasible solution; (**b**) The energy consumption profiles of feasible solution.

## 5.4. Discussions

The model established in this paper integrates the production scheduling, PM planning, and energy controlling for the flow shop under peak demand constraint, which also considers the machine's on/off and speed selection. In Table 1, we can find that the model is unique in the literature. Although it is impossible to directly compare our numerical results with the findings of other references from the viewpoint of quantity, the main conclusions can be compared qualitatively to imply the managerial insights for the practitioners in real industry.

The traditional research about flow shop scheduling problem hard performed good work to search the optimal jobs' sequence; for example, *NEH* method based on the calculation of processing time can provide a solution within a short computation time. However, the assumptions of those references are too ideal and the solution cannot work well in the real plant. The optimal jobs' sequence not only depends on the production requirements but also is strongly affected by the constraints of maintenance planning and energy consideration. According to the findings of our research, the manager needs to

balance the makespan and energy consumption firstly. If the jobs are rush orders, then the manager should choose a high speed level for the machines in order to reduce the makespan and bear the consequence of increasing energy cost. Otherwise, the manager would rather choose a low speed level for the machines than finish the jobs quickly with the sacrifice of energy cost. Different speed levels lead to different processing time for a same job, which will strongly affect the optimal jobs' sequence. For the manager who has decided the relative importance between makespan and energy cost, he/she still needs to solve the integrated model proposed in this paper to find the optimal jobs' sequence since the machines will be shut down when PM is performed.

The execution of PMs should be planned to avoid disturbing the processing of jobs. According to the results in [25], manager should make full use of the idle time between consecutive jobs to perform PMs, which will lead to more PMs than the minimum required number of PMs in order to minimize the makespan. However, the machine needs to be shut down when performing PM, which means that the additional energy needs to be consumed when turning on the machine again. Thus, the manager needs to consider the energy consumption of machines' setup. If it is small, then the suggestion in [25] still works well. Otherwise, the manager should pay attention to the number of PMs in order to reduce the waste of energy. Thus, the impact of PM on the production is not only caused by the unavailability of machine but also via the channel of energy consumption.

The authors of [33–39] investigated when to turn off the machine during the production horizon in order to reduce the energy consumption of idle machine. Let $\gamma$ be the power demand per unit time of idle machine. There is a tie between the consumption of idle machine and the setup consumption, i.e., if the length of idle time is longer than $\tau = setup/\gamma$, the machine should be shut down. Therefore, the arrangement with many small intervals with shorter time is not good; the arrangement with a few large intervals with length $\tau$ is encouraged. The above suggestion is very useful when manager wants to save the energy cost. In addition, we find that the interval of idle time is a good opportunity for performing PMs. If the length of one maintenance is shorter than $\tau$, a PM can be performed here to improve the reliability of machine since it has already been shut down. If the length of one maintenance is longer than $\tau$, then we suggest the manager arrange the production to generate idle intervals whose lengths equal the maintenance time rather than $\tau$.

According to the research results of selecting machines' speed levels to adjust the tradeoff between makespan and energy consumption in literature, it is suggested to set the highest level for each machine in order to finish the jobs as quickly as possible. However, it will lead to a high power demand of the production system. If the peak demand limit is tight, the solution's real performance becomes very bad, as shown in Figure 8. It shows the manager that the makespan cannot be constantly decreased due to the peak demand constraint. Furthermore, the greater the peak limit is, the more the makespan can be reduced. This finding suggests manager to consider the peak limit when evaluating whether a production line can complete an order on time or not. According to the results in this paper, the speed levels of different machines should be matched to guarantee that the maximum power demand is below the limit. Some machines, in which the job's basic processing time is longer, should work with high speed level. The other machines, in which the job's basic processing time is shorter, should work with low speed level. Then, the makespan can be reduced as much as possible under the condition with a good performance in the energy aspect.

## 6. Conclusions

This paper integrates three aspects, namely production, preventive maintenance, and energy consideration for the flow shops, with the PM constraint and peak demand constraint. A mathematical model is established, and a meta-heuristic based on *NSGA-II* and decoding method is designed to solve the model effectively. Using this approach, the Pareto frontier can be obtained to balance the tradeoff between economic objective and environment objective. The numerical results show that impacts of PM constraint and peak demand constraint need to be analyzed case by case.

This research is closely related to the practical application in industrial plant. The main practical implications can be formulated from four points. First, the operations management problem of manufacturing plant is a complicated topic coupled with different aspects. The effectiveness of research will be discounted if researchers only focus on the theoretical study in one aspect. The only way to fill the gap between theory and reality is investigating the integration problem. Second, plant manager can decrease the makespan if more energy consumption can be accepted by the manager. However, the makespan cannot be constantly decreased due to the peak demand constraint. Third, the length of maintenance will strongly affect the makespan and it does not affect the total energy consumption. The positions of PMs should be determined while considering the machines' on/off decision. Fourth, although the total energy consumption is caused by three parts, processing jobs occupies most of energy consumption. Thus, plant manager should pay attention to the processing jobs instead of idle time and setup consumption.

There are several limitations which can be found in the problem assumption. First, we assume that the periodical PM gets rid of the unexpected machine failures and the plant environment is deterministic. Therefore, we establish a deterministic mathematical model in this paper. In the future, the impact of uncertain factors can be considered, e.g., the unexpected machine failures, the rush orders, and the temporary blackout of electricity. Second, we use the total energy consumption as the energy objective. Therefore, it is unnecessary to insert idle times into the schedule to adjust its consumption pattern. In the future, the time-of-use tariff, which provokes the energy users to shift the power demand from peak-hours to valley-hours, can be considered. Third, we assume that there is only one machine in each stage of flow shop. Therefore, the machine allocation problem is avoided in this paper. In the future, parallel machines in each stage can be considered to compose the hybrid flow shop. In addition, only one energy resource is considered in this paper. Renewable energy resources such as solar energy and wind power are encouraged to be alternatives of fossil fuels since renewable resources are cleaner and greener. In the future, it is also very interesting to investigate how to supply the power demand of plant using the micro-grid system.

**Author Contributions:** Conceptualization, W.C. and B.L.; Methodology, W.C. and B.L.; Software, W.C.; Validation, W.C.; and Writing—original draft, W.C. All authors have read and agreed to the published version of the manuscript.

**Funding:** This research was funded by National Natural Science Foundation of China, grant number 71801147, and Shanghai Pujiang Program.

**Conflicts of Interest:** The authors declare no conflict of interest. The funders had no role in the design of the study; in the collection, analyses, or interpretation of data; in the writing of the manuscript, or in the decision to publish the results.

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
