# Peer review of "A Bi-Objective Approach to Minimize Makespan and Energy Consumption in Flow Shops with Peak Demand Constraint"

_sustainability, doi:10.3390/su12104110_

Round 1

Reviewer 1 Report

The Authors present an integrated model consisting of production scheduling, preventive maintenance (PM) planning, and energy controlling applied to a flow shops with the PM constraint and peak demand constraint. The machine’s on/off and the speed level selection are considered to save the energy consumption in this problem. In order to minimize the makespan and the total energy consumption simultaneously, a multi-objective algorithm found on NSGA-II is designed to solve the model effectively. The topic is really interesting, but some parts of the paper are critical.

Following, I report the main points:

  • I suggest inserting in the introduction chapter a table where this research is compared with the others considered in Literature and where you explain the main differences
  • How do you define Etj? How do you calculate it?
  • Lingo has the possibility to solve non-linear problems; do you have tried?
  • Are the parameters of the GA model correct? I suggest you presenting or explaining how you have defined the values of these parameters.
  • I think that you have to enrich the part about the managerial insights in order to increase the value of the work.

Other points:

  • Row 282: you have to mention these researches.
  • Page 11: the font of the graph is too big.
  • For Figure 5-a and figure 6-b you have to redefine the scale.

Reviewer 2 Report

The paper deals with energy efficiency solutions by developing new advanced tools like integrated model consisting of production scheduling, preventive maintenance (PM) planning, and energy controlling in flow shops. The paper is well prepared and deals topical issues however it should be improved. The paper lacks discussion section. All results and findings should be discussed and compared with other studies analysed in literature review section. The conclusions are pour the practical implications should be clearly formulated. The limitations of this study should be properly addressed and future research guidelines should be formulated based on identified limits. Therefore, the paper need major revision. Also methodology should be more clearly presented by providing theoretical  background for the methodology established.

Round 2

Reviewer 1 Report

The authors adjusted the paper by using the suggestions presented in the previous referee. No other notes about the work.

Reviewer 2 Report

The authors have improved their paper and addressed all my comments. They also provided comprehensive answers to my comments. The paper can be published in current form according to my opinion.